# LLM-HFR-RL: Large Language Model (LLM)-Driven Cross-Modal Fine-Grained Alignment and Reinforcement Learning for the Prediction of Heart Failure Risk

## Abstract

Predicting Heart Failure Risk (HFR) using electronic health records (EHR) and generating actionable clinical decisions face significant challenges, including integrating multimodal data, modeling longitudinal temporal patterns, and translating predictions into executable interventions. To address these limitations, this paper proposes the LLM-HFR-RL framework, bridging the gap from risk prediction to clinical decision-making. This framework integrates three key technical innovations: (1) a longitudinal laboratory index summarization method leveraging large language models (LLMs), which transforms discrete test value sequences into clinically meaningful trend summaries; (2) a ternary cross-modal fine-alignment architecture that integrates semantic representations across structured test sequences, LLM-generated trend summaries, and clinical text; and (3) the novel integration of a Reinforcement Learning (RL)-driven decision engine, which learns optimal testing strategies via a multi-objective reward function to dynamically refine clinical decisions. Experimental results demonstrate that LLM-HFR-RL not only significantly improves HFR prediction performance but also forms a high-precision and cost-effective clinical decision support system, providing a new paradigm for intelligent medical intervention.

## 1 Introduction

Cardiovascular diseases, particularly HF, represent a pressing global public health challenge due to their rising prevalence and substantial socioeconomic burden (Groenewegen et al., 2020). The precise early identification of high-risk patients is paramount for facilitating timely interventions, optimizing healthcare resource allocation, and enhancing patient outcomes. EHR systems amass vast amounts of multifaceted patient data, encompassing demographics, vital signs, laboratory results, medication histories, diagnostic codes, imaging reports, and clinical records; thereby providing a comprehensive data source for AI-driven HF risk prediction. While deep learning-based risk prediction models have advanced considerably in the healthcare domain in recent years (Yoon et al., 2023), researchers however encounter two interconnected core challenges when leveraging EHR data for accurate HF risk stratification: the effective integration of heterogeneous multi-modal data and the robust modeling of patients' longitudinal temporal trajectories. Moreover, achieving accurate risk prediction alone is insufficient to support clinical practice: translating these predictions into actionable intervention suggestions that balance effectiveness and cost poses another major challenge for intelligent medical decision support systems.

On one hand, EHR data inherently represents a highly heterogeneous multimodal data source. A significant semantic gap exists between structured data (e.g., patient diagnoses, medication records) and unstructured data (e.g., clinical notes, discharge summaries). Conventional modeling approaches (Zhao et al., 2023; Naseem et al., 2024; Zhang et al., 2022) often rely on simple fusion techniques (e.g., early or late fusion) or extract only superficial features across these modalities, which limits their ability to capture the profound clinical interrelations and complementary information. This results in fragmented multimodal semantics, ultimately constraining the model's capacity for comprehensive health profiling. On the other hand, disease pathogenesis—particularly for chronic con-

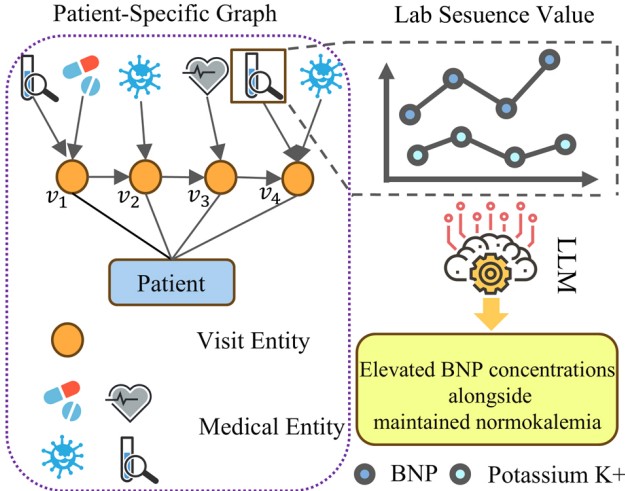

Figure 1: Example: Modeling longitudinal temporal data into a personalized patient graph composed of four medical entity types and patient visit entities, and constructing sequential laboratory test values into trend descriptions.

ditions like HF—manifests as a dynamic progression, such that the temporal evolution patterns of key biomarkers encode crucial pathophysiological significance. As illustrated in Figure 1, our approach leverages LLMs to model temporal laboratory data as interconnected evolutionary patterns of laboratory indicators (such as "elevated BNP concentrations alongside maintained normokalemia").

## 2 RELATED WORK

Research on EHR-based Health Event Prediction can be broadly categorized into several directions, including laboratory metrics, temporal modeling, graph structures, multi-modal fusion, and RL. Early studies predicted events either by manually extracting features from laboratory metrics (Chen et al., 2024; Xu et al., 2022) or utilizing temporal models such as GRU, LSTM, and Transformer (Choi et al., 2017; Liu et al., 2022; Li et al., 2020). While efficient, these approaches are often limited in their ability to capture complex dynamics and structured interdependencies. Graph-based models attempt to introduce external knowledge to construct patient or disease relationship networks (Wu et al., 2023; Kang et al., 2024; Yang et al., 2024). However, they are often constrained by static topologies. Multi-modal methods, despite integrating textual information (Zhao et al., 2023; Lu et al., 2021; Naseem et al., 2024), typically exhibit limited cross-modal interaction mechanisms. Recently, RL has been applied to tasks such as treatment optimization and detection decision-making (Yu et al., 2023; Li et al., 2022). However, these RL-based approaches often lack multi-modal semantics and multi-objective constraints in their state representations and reward function designs.

Although traditional temporal models have laid a foundation for EHR analysis, they often suffer from a lack of structured association and coarse handling of numerical sensitivity. For instance, Choi et al. (2017) employed GRUs to model patient visit sequences, utilizing hidden states to propagate historical information for dynamic heart failure risk prediction. Liu et al. (2022) used LSTMs to model different types of medical events in separate network channels, with gating mechanisms facilitating information exchange between them. Li et al. (2020) introduced the Transformer to EHR analysis, leveraging its self-attention mechanism to capture long-range dependencies across visits and demonstrating superior performance in tasks like in-hospital mortality prediction. Despite their contributions, these methods generally lack structured relational reasoning and exhibit limited numerical sensitivity.

Graphical models more appropriately mine structured knowledge from EHRs by explicitly modeling inter-entity relationships. Wu et al. (2023) leveraged LLMs and external biomedical KGs to generate additional triples when constructing personalized patient graphs. IICL (Kang et al., 2024) captured indirect latent disease relationships among different patients by constructing a static disease

graph based on co-occurring diseases across patients. MMGCN (Yang et al., 2024) built multimodal patient similarity networks using gene expression, copy number variation, and clinical data, integrating them into a unified multi-view network via a similarity network fusion algorithm. However, the rigidity of these static graph structures fails to reflect the temporal dynamics of diagnostic and therapeutic activities.

Many models have begun leveraging textual records in EHRs to enhance predictive performance. Zhao et al. (2023) obtained a global patient representation by applying cross-attention between patient disease and textual data, followed by generating the final patient representation via an attention mechanism. CGL (Lu et al., 2021) derived text representations using TF-IDF corrected attention; however, the patient and text representations were merely summed. Subsequent work, GLLA (Naseem et al., 2024), improved upon CGL by introducing label attention [10] during the word embedding process for text representation. Nevertheless, the integration of patient and text representations remained a simple addition. This approach of directly concatenating feature vectors from different modalities before feeding them into models overlooks the semantic disparities and complex interactions between modalities.

In recent years, RL has demonstrated significant potential in clinical decision support, offering novel pathways for translating predictive models into actionable treatment plans. For instance, EHRs-DQN (Li et al., 2022) adopted a multi-DQN framework inspired by physician consultations to optimize treatment strategies for diabetic patients. SM-DDPO (Yu et al., 2023) employed proximal policy optimization (PPO) to sequentially select laboratory test panels based on historical observations, forming a dynamic decision-making scheme that maintains diagnostic accuracy while reducing testing costs. However, most existing RL methods rely on simplified or handcrafted patient state representations, failing to fully integrate the rich semantic and longitudinal dynamic information available in multimodal EHR data. Moreover, their reward functions are often designed around a single clinical objective, making it challenging to balance multiple real-world constraints such as accuracy, cost, and timeliness in practical applications.

## 3 PROBLEM FORMULATION

**Basic symbols.** The EHR of each patient is represented by a sequence $< v_1, v_2, \cdots, v_T >$ consisting of multiple visits, where $v_t$ represents the $t$-th visit (Kim et al., 2025). For the patient's $t$-th visit $v_t^i = ( c_t^i, m_t^i, l_t^i, d_t^i )$, it contains multiple sets of heterogeneous medical concepts. For improved readability, the patient superscript $i$ will be omitted in the subsequent sections of this paper. For detailed descriptions of the four types of medical concepts, please refer to Appendix A.

We define a Markov Decision Process (MDP) as the tuple $(S, A, P, R, \gamma)$. The state space $S$ represents the patient's health profile, which is generated by a patient state encoder. The action space $A$ is discrete, corresponding to the model's selection of one laboratory test from $|K|$ available options. The transition function $P$ defines the probability of moving to a new state $s_{t+1}$ after taking action $a_t$ in state $s_t$. The reward function $R$ is a sophisticated multi-objective function. The discount factor $\gamma \in [0, 1]$ balances the importance of immediate and future rewards.

**Heart failure risk prediction.** The primary objective of this study is to learn a prediction function $f$ that estimates the probability of a patient developing heart failure within a specific future time window, based on their complete historical records up to the t-th visit.

**Optimal clinical decision strategy learning.** Building upon accurate risk prediction, this study further aims to provide decision support for clinical intervention via joint learning of both tasks. This objective is formulated as a MDP and addressed within a RL framework.

## 4 METHODOLOGY

We propose an end-to-end multimodal longitudinal fusion framework, whose core idea is to jointly learn a unified architecture capable of both accurate risk prediction and optimal testing strategy generation through the co-optimization of predictive loss and policy gradients. An overview of the entire pipeline is illustrated in Figure 2.

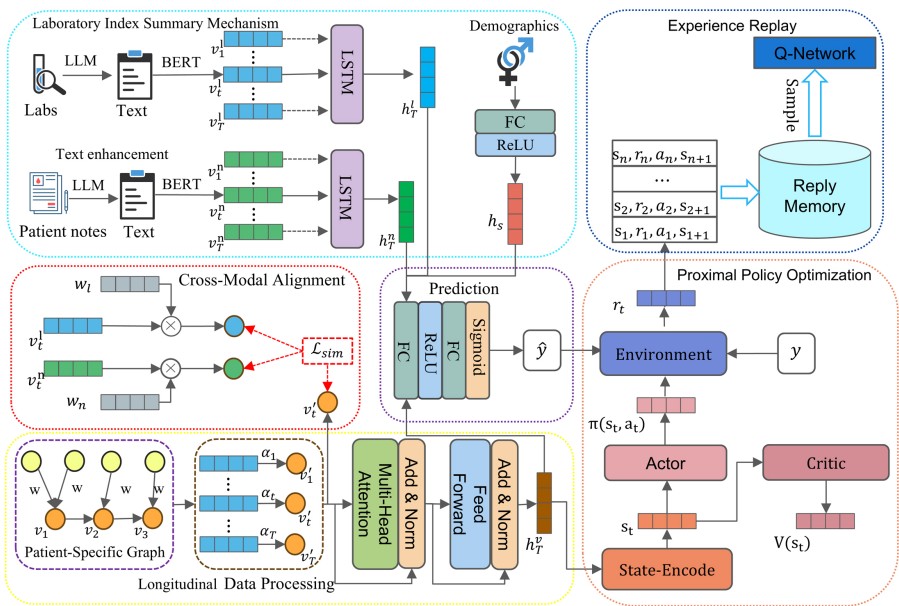

Figure 2: Overall Framework of the Model. The proposed framework primarily incorporates the following methodologies: LLM-driven information enhancement, longitudinal temporal modeling of EHR, joint RL, and multi-modal alignment.

## 4.1 INPUT EMBEDDING AND REPRESENTATION

**LLM-powered embeddings.** First, for each laboratory indicator $k \in 1, 2, \cdots, |K|$, we extract its non-uniform temporal subsequence $l_{t,k} = (med_k, (v_h, \tau_h)_{h=1}^{H_k})$. Here, $v_h$ denotes the measured value, $\tau_h$ is the timestamp in hours, $med_k$ is the medical term extracted from clinical corpora corresponding to the k-th indicator, and $H_k$ represents the number of measurements for indicator $k$ during the current patient visit. Using the powerful in-context learning capability and extensive domain knowledge of the DeepSeek model (Guo et al., 2025), we achieve an intelligent transformation from the raw measurement sequence $l_t$ to a textual summary $z_t$ capturing temporal trends. This approach transcends the limitations of conventional numerical analysis.

$$z_t^l = LLM(l_t, \theta_{LLM}, x_l, \mathcal{T}) \tag{1}$$

where $\theta_{LLM}$ is the parameter of knowledge of the frozen model, $\mathcal{T}$ is the temperature coefficient, and $x_l$ is the word cue. The generated trend text $z_t^l$ was converted into a dense semantic vector using the Bio_ClinicalBERT model (Alsentzer et al., 2019), which was pre-trained on the entire clinical notes of the MIMIC-III (Johnson et al., 2016) database.

$$H^{last} = BERT(z_t^l, \theta_B) \tag{2}$$

$$v_t^l = \frac{1}{L} \sum_{i=1}^{L} h_i^{last} \tag{3}$$

where $\theta_B$ denotes the pre-trained parameters, $H^{last}$ is the last hidden state, $L$ is the number of tokens, and $v_t^l$ is the mean-pooled visit-level patient state representation. Since medical decision-making requires the comprehensive integration of a patient's historical states, we employ an LSTM to model the sequence of clinical trend semantic vectors $< v_1^l, v_2^l, \cdots, v_T^l >$ generated from laboratory tests:

$$h_t^l = LSTM(v_t^l, h_{t-1}^l) \tag{4}$$

where $h_t^l$ denotes the hidden state at the t-th visit. Finally, the hidden state of the last visit $(t = T)$, denoted $h_T^l$, is taken as a history-aware representation of the laboratory trend text, which integrates all historical information.

Patient clinical records serve as the central repository of the diagnostic and therapeutic process, whose quality directly impacts the accuracy and continuity of medical decision-making. However, they often face challenges such as information fragmentation, spelling errors, and terminology inconsistency. This study leverages LLMs to perform semantic enhancement and structural reorganization of raw physician notes:

$$z_t^n = LLM(n_t, \theta_{LLM}, x_n, \mathcal{T}) \tag{5}$$

where $x_n$ is the prompt, the temperature coefficient is similarly set to $\mathcal{T} = 0.1$, and $n_t$ denotes the patient's clinical notes. Consistent with the LLM-powered temporal summarization of laboratory metrics, we first obtain a dense semantic vector of the enhanced clinical note using Bio_ClinicalBERT. This vector sequence is then modeled by an LSTM:

$$N^{last} = BERT(z_t^n, \theta_B) \tag{6}$$

$$v_t^n = \frac{1}{L} \sum_{i=1}^{L} n_i^{last} \tag{7}$$

$$h_t^n = LSTM(v_t^n, h_{t-1}^n) \tag{8}$$

where $h_t^n$ represents the hidden state of the t-th visit. The hidden state of the final visit, $h_T^n$, is taken as the history-aware representation of the clinical notes, integrating all historical information.

**Longitudinal data modeling and embedding.** The patient's longitudinal data are structured into a dynamic, time-decayed weighted, and patient-personalized KG, denoted as $\mathcal{G} = (\mathcal{V}, \mathcal{L}, \mathcal{W})$. For the detailed construction process of the personalized patient graph, please refer to Appendix B. We initialize a node embedding matrix $E \in \mathbb{R}^{|V| \times d}$, where $|\mathcal{V}|$ is the node cardinality and $d$ the embedding dimension. Initial embedding vectors $e$ are subsequently retrieved as follows:

$$e = E(V) \tag{9}$$

For example, to perform information aggregation for a node $e_i \in \mathbb{R}^d$ in graph $\mathcal{G}$, the embedding update formula for the $(l + 1)$-th layer is as follows:

$$e_i^{l+1} = \sigma\left( \sum_{k \in \mathcal{N}(v_i)} w_{k,i} \cdot e_k^l \right) \tag{10}$$

where $\mathcal{N}(v_i)$ denotes the set of neighboring nodes of node $i$, $w_{k,i} \in \mathbb{R}$ is the edge weight from node $k$ to node $i$, and $\sigma$ is the $ReLU$ activation function. The visit embedding $v_t^L \in \mathbb{R}^d$ is obtained through $L$ layers of graph convolution. Since Kim et al. (2024) observed that not all historical information is beneficial, this study leverages the laboratory trend text to compute an attention score for each visit:

$$v_t' = v_t^L \odot (W_a^T v_t^l + b_a) \tag{11}$$

where $W_a^T \in \mathbb{R}^{d \times 1}$ and $b_a \in \mathbb{R}$ are learnable parameters, and $\odot$ denotes the element-wise product. To capture long-term dependencies in the patient visit sequence, the graph embeddings from the previous T visit nodes are concatenated in chronological order into a sequence $V' = v_1', v_2', \cdots, v_T'$, which is then fed into a Transformer encoder:

$$[h_1^v, h_2^v, \cdots, h_T^v] = Encoder(V') \tag{12}$$

Finally, the embedding of the final visit $h_T^v$, output from the last layer of the Transformer, is taken as the globally attentive patient historical state representation. This representation achieves hierarchical

extraction of spatiotemporal features. Simultaneously, this state vector $h_T^v$ will also serve as the input to the RL agent.

In the demographic embedding, continuous variables are standardized via Z-score normalization, while categorical variables are converted into sparse vectors using multi-hot encoding. The encoded demographic sparse vector $f \in \mathbb{R}^{d_f}$ is then mapped to a low-dimensional dense vector $h_s \in \mathbb{R}^{d_s}$ through an embedding layer, as formulated by:

$$h_s = \mathcal{W}_s \cdot f^T + b_s \tag{13}$$

## 4.2 SUPERVISED PREDICTION HEAD

To provide stable training signals, we employ an MLP classifier for endpoint prediction based on the fused state of the three history-aware representations—$h_T^n$, $h_T^l$, and $h_T^v$—and the demographic embedding $h_s$:

$$h_z = cat\left(h_T^n, h_T^l, h_T^v, h_s\right) \tag{14}$$

$$\hat{y} = Sigmoid(\mathcal{W}_2 \cdot ReLU\left(\mathcal{W}_1 \cdots h_f + b_1\right) + b_2) \tag{15}$$

where a *dropout* layer is applied to the hidden layer to mitigate overfitting; $h_z \in \mathbb{R}^{d_z}$ is the concatenated feature vector; $\mathcal{W}_1 \in \mathbb{R}^{d_z \times d_h}$ and $\mathcal{W}_2 \in \mathbb{R}^{d_h \times 1}$ are learnable parameters; $d_h$ denotes the hidden layer dimension; and the $Sigmoid$ function compresses the output to the interval $[0, 1]$, representing the probability of HF occurrence. The supervised loss is the binary cross-entropy loss:

$$\mathcal{L}_{SL} = -\frac{1}{N} \sum_{i=1}^{N} y_i log\widehat{y_i} + (1 - y_i)log(1 - \widehat{y_i}) \tag{16}$$

## 4.3 REINFORCEMENT LEARNING AGENT

We employ an Actor-Critic architecture as our RL agent. The global attentive patient history state representation, $h_T^v$, is transformed into a patient state, $s$, via a state encoder.

$$s = \mathcal{W}_s \cdot f^T + b_s \tag{17}$$

The actor network is a policy network that maps a state $s_t$ to a probability distribution over the action space. The agent then samples an action $a_t \sim \pi(\cdot|s_t)$ from this distribution. The critic network is a value network designed to estimate the expected cumulative return of a state $s_t$.

$$\pi(a_t \mid s_t; \theta_a) = \text{softmax}(\mathbf{W}_{a2} \cdot \text{ReLU}(\mathbf{W}_{a1}s_t + \mathbf{b}_{a1}) + \mathbf{b}_{a2}) \tag{18}$$

$$V(s_t; \theta_c) = \mathbf{W}_{c2} \cdot \text{ReLU}(\mathbf{W}_{c1}s_t + \mathbf{b}_{c1}) + \mathbf{b}_{c2} \tag{19}$$

The reward function is crucial for the success of RL. A composite reward function—integrating clinical accuracy, cost-effectiveness, and the timing of intervention—is designed as follows:

$$R(s_t, a_t) = R_{acc} + R_{cos} + R_{early} \tag{20}$$

where $R_{acc} = 1 - |\hat{y} - y|$ denotes the accuracy reward, which incentivizes correct risk prediction; $R_{cos} = -C(a_t)$ represents the cost penalty, encouraging the selection of cost-effective tests to achieve effective control of detection costs; and $R_{early} = \mathbb{I}(y = 1 \wedge \hat{y} > 0.7)$ provides a substantial reward for successfully identifying high-risk patients and intervening early.

## 4.4 Joint Optimization Training Algorithm

We introduce a hybrid training strategy to collaboratively optimize the supervised learning loss and the RL objective.

**Proximal policy optimization (PPO).** For the policy component, the PPO algorithm is employed to update the actor and critic networks, with the following objective function:

$$\mathcal{L}^{\text{PPO}}(\theta) = \mathbb{E}_t[r_t(\theta)\widehat{A_t}, \text{clip}(r_t(\theta), 1-\epsilon, 1+\epsilon)\widehat{A_t}] - c_1(V_\theta(s_t) - V_{\text{target}})^2 + c_2\mathcal{H}(\pi_\theta(\cdot \mid s_t)) \quad (21)$$

where $\theta$ denotes the parameters of the policy network and the value network, $r_t(\theta) = \pi_\theta(a_t|s_t)/\pi_{\theta_{old}}(a_t|s_t)$ is the probability ratio of the new policy versus the old policy for taking an action, $\widehat{A_t}$ represents the Generalized Advantage Estimate (GAE), $V_\theta(s_t)$ is the state-value estimate from the Critic network, $V_t^{targ}$ denotes the target value, and $\mathcal{H}$ is the policy entropy term which encourages exploration by the model.

To further enhance the stability of value estimation, we introduce an auxiliary Q-network and an experience replay mechanism. The experience replay buffer stores experience tuples—denoted as $(s_t, a_t, r_t, s_{t+1})$—generated from the agent's interaction with the environment in a fixed-size replay buffer, $\mathcal{D}$. During the Q-network update, a batch of experiences, $\mathcal{B} \sim \mathcal{D}$, is randomly sampled from this buffer. The target value for the Q-network is computed using a target network:

$$y = r + Q_{\theta_{\text{target}}}(s_{t+1}, \mu(s_{t+1})) \quad (22)$$

where $\mu(s_{t+1}) = \arg\max_a Q_\theta(s_{t+1}, a)$. The parameters of the target Q-network are synchronized with the online network via a soft update: $\theta_{target} \leftarrow \tau\theta + (1-\tau)\theta_{target}$. The loss function for the Q-network is defined as:

$$\mathcal{L}_{QL}(\theta) = \frac{1}{|\mathcal{B}|} \sum_{(s,a,r,s')\in\mathcal{B}} (Q_\theta(s,a) - y)^2 \quad (23)$$

**Cross-modal alignment.** Given that text embeddings and visit-level graph embeddings reside in distinct vector spaces, this mechanism first projects the text embeddings into the graph embedding space. It then establishes visit-level triple-wise cross-modal alignment by minimizing a similarity-based loss. We note the perspective from Kim et al. (2025): even across different patients, their visit embeddings can be valuable (i.e., cross-patient similarity exists). Accordingly, while our method acknowledges this insight, it does not explicitly maximize the similarity between different patients or visits.

$$\mathcal{L}_l = \frac{1}{N} \sum_{i=1}^{N} \frac{1}{T} \sum_{t=1}^{T} (1 - \cos(v'_t, W_l^T \cdot v_t^l)) \quad (24)$$

$$\mathcal{L}_n = \frac{1}{N} \sum_{i=1}^{N} \frac{1}{T} \sum_{t=1}^{T} (1 - \cos(v'_t, W_n^T \cdot v_t^n)) \quad (25)$$

$$\mathcal{L}_{sim} = \mathcal{L}_n + \mathcal{E}\mathcal{L}_l \quad (26)$$

where $\cos(\cdot)$ denotes the cosine similarity function, whose value range is $[-1, 1]$. Consequently, the range of $(1 - \cos(\cdot))$ is $[0, 2]$. Here, $W_l^T$ is a projection matrix (where the superscript $T$ denotes transpose), $N$ is the number of patients, and $T$ is the maximum number of visits.

**Overall training objective.** The final training objective of the model is formulated as a weighted sum of the supervised learning loss and the RL loss:

$$\mathcal{L}_{\text{Total}} = \mathcal{L}_{\text{sim}} + \mathcal{L}_{\text{SL}} + \beta\mathcal{L}_{\text{PPO}} + \mathcal{L}_{\text{QL}} \quad (27)$$

Table 1: HF Prediction Results Using AUC (%), F1 (%), and ACC (%) on the MIMIC-III and e-ICU Datasets. The best performers are in bold.

| Models | MIMIC-III | | | eICU | | |
|---|---|---|---|---|---|---|
| | AUC | F1-Score | ACC | AUC | F1-Score | ACC |
| LSTM | 84.4 | 68.4 | 77.8 | 88.6 | 78.9 | 92.9 |
| Transformer | 81.6 | 70.7 | 76.9 | 88.9 | 79.5 | 94.2 |
| Retain | 77.5 | 65.2 | 78.0 | 88.7 | 81.8 | 94.1 |
| Timeline | 82.3 | 71.0 | 80.2 | 89.0 | 79.6 | 93.7 |
| CGL | 82.5 | 71.1 | 79.3 | 90.6 | 80.1 | 93.8 |
| Chet | 84.9 | 71.6 | 79.8 | 86.7 | 72.9 | 92.8 |
| tBNA-PR | 85.2 | 72.4 | 79.7 | 91.9 | 81.6 | 94.6 |
| HAR-LSTM | 84.7 | 73.1 | 79.1 | 89.1 | 79.8 | 94.0 |
| SHY | 84.5 | 71.2 | 79.8 | 90.1 | 80.7 | 94.3 |
| LLM-HFR-RL | **88.3** | **76.4** | **83.2** | **93.9** | **83.7** | **95.2** |

All parameters are jointly optimized via gradient descent, thereby enabling the model to simultaneously acquire the capability for accurate patient risk prediction and the generation of cost-effective clinical decision-making strategies.

## 5 EXPERIMENTS

**Datasets.** The study utilizes large-scale datasets derived from MIMIC-III (Johnson et al., 2016) and eICU (Pollard et al., 2018). Patients with fewer than two visit records were excluded. We extracted relevant medical entities including diseases, medications, laboratory tests, vital signs, and clinical notes. Disease codes are standardized using Level 3 of the modern ICD-9-CM classification system through pre-definition. Notably, while eICU lacks physician progress notes, it provides comprehensive past medical history documentation. For further details regarding the dataset, please see Appendix B. A stratified sampling strategy partitioned patients into training, validation, and test sets at an 7:1.5:1.5 ratio.

**Implementation details.** Network architecture hyperparameters were kept constant throughout the study. Models were implemented in PyTorch and trained on NVIDIA GeForce RTX 3090 GPUs. Dataset splits were randomized using fixed seeds. The embedding layer dimension was set to 512, with dropout applied to hidden layers to prevent overfitting. Optimization employed Adam (Choi et al., 2016) with an initial learning rate of 1e-3. The training process begins with a pre-training phase on the supervised task, followed by a joint training phase that integrates the pre-trained model with RL. For further details on hyperparameters, please refer to Appendix B.Following Chet (Lu et al., 2022), we adopt AUC and F1-score as primary metrics, with accuracy (ACC) included as a supplementary measure.

**Comparison algorithms.** To comprehensively evaluate our work, LLM-HFR-RL was benchmarked against: Two classical sequence models (spec.,LSTM (Graves, 2012) and Transformer (Vaswani et al., 2017)) and seven state-of-the-art clinical prediction methods (spec. RETAIN (Choi et al., 2016), Chet (Lu et al., 2022), HAR-LSTM (Wang et al., 2024), SHy (Yu et al., 2025), CGL (Lu et al., 2021), tBNA-PR (Liang & Guo, 2023), and Timeline (Bai et al., 2018)).

### 5.1 RESULTS AND ANALYSIS

**Ablation study.** To rigorously validate the contribution of each module within the proposed framework, a series of ablation studies were conducted. The evaluated variants included: removing the trend text summaries of laboratory tests (Without Lab Text), ablating the patient notes (Without Notes Text), excluding all textual information (Without Full Text), disabling the triplet alignment loss (Without $L_{sim}$), removing the PPO algorithm for reinforcement learning (Without PPO), and ablating the Q-learning component with experience replay (Without Q-Learning). As illustrated in Table 3, the removal of either LLM-driven laboratory summaries or patient notes resulted in a significant performance drop, particularly in the F1-score, underscoring the critical importance of deep multi-modal integration for accurate identification of positive cases. Furthermore, compared to a

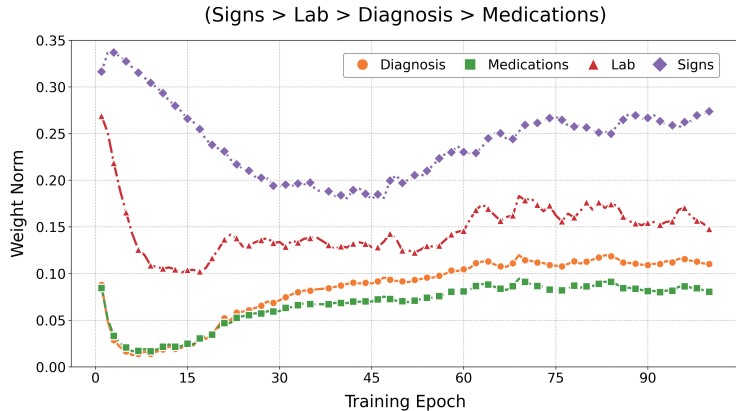

Figure 3: Evolutionary trajectory of medical entity node weights across training epochs.

supervised learning-only baseline, our full framework integrating the RL decision engine yielded significant improvements in both AUC and F1-score, highlighting the substantial contribution of the joint optimization strategy.

**Visual analytics.** As illustrated in Figure 3, the model weight visualization reveals distinct weighting patterns across clinical entity types. Notably, signs and laboratory findings demonstrate the highest predictive influence for HF, validating the core principle of evidence-based medicine: data-driven diagnosis. All four clinical entity types exhibit consistent weight dynamics—an initial sharp decline followed by gradual resurgence. This trajectory indicates the model's adaptive learning process regarding feature dependencies, with the inflection point signifying discovery of inter-feature synergies. Crucially, diagnoses and medications display steadily increasing weights with strong mutual correlation, revealing the intrinsic clinical logic: precise diagnostics drive optimal therapeutic decisions.

## 6 CONCLUSION

The LLM-HFR-RL framework successfully addresses core challenges in HFR prediction—including dynamic temporal modeling, heterogeneous data fusion, and clinical decision support—through the organic integration of LLM-enhanced multimodal RL and RL-based decision optimization. Experimental results robustly validate its exceptional performance and generalization capability. However, certain limitations remain. First, the action space of the RL component is currently limited to laboratory tests; future work could extend it to include a broader range of clinical interventions such as medication therapies and imaging studies. Second, the weighting scheme of the reward function requires further refinement. Exploring inverse learning from clinical outcomes or developing adaptive reward functions represents an important direction for future research.

ACKNOWLEDGMENTS

This work was financially aided by the Natural Science Foundation of China(61976074).

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

## LLM USAGE DISCLOSURE

In this work, LLMs were utilized as auxiliary efficiency tools to assist the author strictly limited to generating code annotations based on the author's instructions. These AI-assisted materials do not possess independent scholarly authority. The core intellectual contributions of this research—including the research design, key insights, data interpretation, rigorous validation, and final content approval—were entirely conducted by the author. The author has thoroughly reviewed, corrected, and where necessary, rewritten all LLM-generated content to ensure its accuracy and scientific validity. The author assumes full responsibility for every academic claim, line of code, and all research findings presented in this manuscript. This research strictly adheres to academic ethics.

This statement is provided as required.

## A  PROBLEM FORMULATION

### A.1  CLINICAL CONCEPT

The diagnosis is represented by a multi-hot binary vector $c_t^i \in \{0,1\}^{|C|}$, and the medication by $m_t^i \in \{0,1\}^{|M|}$, where $|C|$ and $|M|$ denote the total number of diagnosis and medication codes,

respectively. A value of 1 at a specific position indicates that the corresponding diagnosis or medication is present in the visit, and 0 otherwise. Since the number and type of laboratory tests and vital signs measurements may vary across visits, laboratory test results $l_t^i \in \mathbb{R}^{|K| \times |H|}$ and vital signs $d_t^i \in \mathbb{R}^{|F| \times |J|}$ are represented as real-valued matrices. Here, $|K|$ and $|F|$ are the total number of laboratory test types and vital sign types (fixed dimensions), while $|H|$ and $|J|$ represent the number of measurements for laboratory tests and vital signs in the current visit (variable dimensions). Additionally, $n_t^i$ denotes the clinical note written by the physician during the visit.

## A.2 PATIENT-SPECIFIC GRAPH

The node set $\mathcal{V}$ comprises patient visit nodes $v_t$ and four types of medical entity nodes: diagnoses $c$, medications $m$, laboratory tests $l$, and symptoms $d$. Since $l$ and $d$ may have multiple measurements within a single visit, this study retains only the first measured value. The edge set $\mathcal{L}$ contains directed edges from medical entity nodes $(c, m, l, d)$ to their corresponding visit node $v_t$, as well as temporal progression edges $v_{t-1} \rightarrow v_t$ connecting consecutive visits. The time-decayed edge weights $\mathcal{W}$ for laboratory tests $(l)$ and symptoms $(d)$ are defined as the product of the normalized original value and the inverse of the time interval between adjacent visits, i.e., $W = \frac{1}{\triangle_{t,t+1} + \epsilon} \times x_{norm}$, where $\triangle t, t+1$ is the time difference between adjacent visits and $\epsilon$ is a small constant to prevent division by zero. The weights for all other edges are directly defined by the inverse time interval ($W = \frac{1}{\triangle_{t,t+1} + \epsilon}$). Specifically, the time interval for the last visit node $v_T$ to the four types of medical concepts is set to 1, thereby enhancing the recency weight of the most recent visit information.

# B DATASETS AND HYPERPARAMETERS

The statistical information of the datasets is presented in Table 2, and the detailed hyperparameters are summarized in Table 3 and 4.

Table 2: Dataset statistics on MIMIC-III and eICU

| Dataset | MIMIC-III | eICU |
|---|---|---|
| # patients | 7,537 | 11,691 |
| HF. # patients | 2,659(25.3%) | 1,731(14.8%) |
| Max. # visit | 41 | 8 |
| # entitys | 15,670 | 15,552 |
| # edges | 1,634,654 | 689,074 |

Table 3: Hyperparameters on MIMIC-III and eICU. Joint training: The starting epoch for joint training.

| Hyperparameters | MIMIC-III | eICU |
|---|---|---|
| $\beta$ | 0.3 | 0.3 |
| $\mathcal{E}$ | 0.5 | 0.5 |
| Batch size | 256 | 256 |
| Embedding dimension | 512 | 256 |
| lr | 0.001 | 0.001 |
| rl_lr | 0.0003 | 0.0003 |
| $\epsilon$ | 0.2 | 0.2 |
| $c_1$ | -0.5 | -0.5 |
| $c_2$ | -0.01 | -0.01 |
| $\gamma$ | 0.99 | 0.99 |
| Action space | 29 | 29 |
| Joint training | 5 | 5 |

Table 4: Cost weights of laboratory test items in the reinforcement learning action space

| Specific Test Items | Cost Category | Cost Weight |
|---|---|---|
| Hemoglobin | Basic Tests | 0.1 |
| Hematocrit | Basic Tests | 0.1 |
| WBC | Basic Tests | 0.1 |
| Platelets | Basic Tests | 0.1 |
| RBC | Basic Tests | 0.1 |
| Na | Basic Tests | 0.1 |
| K | Basic Tests | 0.1 |
| Ca | Basic Tests | 0.1 |
| Mg | Basic Tests | 0.1 |
| BUN | Basic Tests | 0.1 |
| Creatinine | Basic Tests | 0.1 |
| Glucose | Basic Tests | 0.1 |
| Total Protein Urine | Advanced Tests | 0.3 |
| Urine Glucose | Advanced Tests | 0.3 |
| Urine RBC | Advanced Tests | 0.3 |
| Cholesterol | Advanced Tests | 0.3 |
| LDL | Advanced Tests | 0.3 |
| HDL | Advanced Tests | 0.3 |
| Triglycerides | Advanced Tests | 0.3 |
| ALT | Advanced Tests | 0.3 |
| AST | Advanced Tests | 0.3 |
| Bilirubin | Advanced Tests | 0.3 |
| Albumin | Specialized Tests | 0.5 |
| ALP | Specialized Tests | 0.5 |
| Iron | Specialized Tests | 0.5 |
| Ferritin | Specialized Tests | 0.5 |
| TIBC | Specialized Tests | 0.5 |
| TSH | Specialized Tests | 0.5 |
| NT-proBNP | Specialized Tests | 0.5 |

