# OpenReview forum: "LLM-HFR-RL: Large Language Model (LLM)-Driven Cross-Modal Fine-Grained Alignment and Reinforcement Learning for the Prediction of Heart Failure Risk"
_ICLR.cc/2026/Conference — Submitted to ICLR 2026_

### Official Review · Reviewer_L5KX · 2025-10-31

**Soundness:** 2
**Presentation:** 2
**Contribution:** 2
**Rating:** 2
**Confidence:** 5

**Summary:**

This paper presents the LLM-HFR-RL framework for heart failure risk (HFR) prediction using electronic health records (EHR). The proposed system combines three main technical innovations: (1) leveraging large language models (LLMs) for longitudinal laboratory data summarization; (2) implementing a ternary cross-modal fine-alignment network to fuse structured sequences, LLM-generated clinical trend summaries, and text records; and (3) incorporating a reinforcement learning (RL) agent for learning optimal, cost-sensitive clinical testing strategies. The framework is evaluated on the MIMIC-III and eICU datasets, demonstrating improvements over various state-of-the-art baselines and ablations.

**Strengths:**

1. Integrated Multimodal Architecture: The paper effectively unites structured EHR components, unstructured clinical notes, and LLM-powered temporal laboratory trend summaries. The ternary fusion approach enables richer patient modeling compared to simple concatenation or late fusion techniques.

2. RL-Driven Decision Engine: Explicitly models the process from risk prediction to actionable, cost-aware clinical decision-making via a well-motivated multi-objective RL paradigm. The reward function is designed to jointly consider prediction accuracy, testing costs, and timeliness of intervention.

**Weaknesses:**

1. While Table 1 (Page 8) summarizes main comparative performance, there is no discussion of how hyperparameters were tuned or controlled across baselines. The lack of confidence intervals, significance tests, or statistical reporting weakens the reliability of claimed improvements (e.g., is 0.5% AUC increase statistically meaningful?). Furthermore, it is unclear if all baselines were reimplemented or taken from published numbers; this distinction matters for fair comparison.

2. Details Missing for RL Training and Stability: The integration of both PPO and a Q-learning auxiliary head appears innovative (Sections 4.3–4.4), but more justification is needed as to why combining two value-based RL approaches stabilizes training. No ablation or figure is presented to compare PPO-only, Q-learning only, and combined variants. There is also a lack of information on convergence criteria, reward clipping, or hyperparameter tuning strategies, which could impact reliability, especially on small/imbalanced medical datasets.

3. Motivation for RL is unclear. The paper does not clearly justify why reinforcement learning is needed, since the final objective still relies on supervised prediction metrics (AUC/F1). The cost reward appears manually assigned rather than data-driven, and it is not explained how these cost labels were obtained or validated. Without showing that the RL policy meaningfully changes testing behavior or efficiency, the use of RL seems conceptually weak and possibly unnecessary.

4. Redundant Discussion of RL-Related Work. The Related Work section repeats reinforcement learning (RL) studies twice—first as part of the general categorization of EHR modeling directions and again in a separate paragraph that restates the same references. This redundancy weakens the narrative flow and makes the section appear unfocused.

5. Insufficient comparison with recent LLM-based approaches. The experimental evaluation mainly contrasts the proposed framework with traditional EHR models (e.g., LSTM, RETAIN, CGL), but omits comparisons with more recent LLM-driven or retrieval-augmented methods such as GraphCare (2024), DearLLM (2025), EMERGE(2024) and RAM-EHR (2025). These methods represent the current state of LLM-based medical reasoning and multimodal alignment. Without benchmarking against such contemporaries, it is difficult to assess whether LLM-HFR-RL provides genuine advances over the latest generation of LLM-enhanced EHR frameworks.

6. Limited contribution from the LLM component. Although the framework leverages a large language model to summarize laboratory trends, the LLM’s role appears superficial—serving mainly as a text generator rather than an active reasoning or representation-learning component. The overall architecture largely combines existing encoding modules (BERT, LSTM, Transformer, RL) in a structured pipeline, giving the impression of a well-engineered composition rather than a fundamentally new integration of LLM capabilities into clinical reasoning or temporal modeling.

Ref:

GraphCare: Enhancing Healthcare Predictions with Personalized Knowledge Graphs

DearLLM: Enhancing Personalized Healthcare via Large Language Models-Deduced Feature Correlations

EMERGE: Integrating RAG for Improved Multimodal EHR Predictive Modeling

RAM-EHR: Retrieval Augmentation Meets Clinical Predictions on Electronic Health Records

**Questions:**

See Weaknesses.

---

### Official Review · Reviewer_QkjM · 2025-11-01

**Soundness:** 2
**Presentation:** 3
**Contribution:** 2
**Rating:** 2
**Confidence:** 3

**Summary:**

The paper introduces an EHR temporal prediction model, integrating LLM temporal reasoning and graph analysis for multimodal data representation. The paper then augments the Deep Learning architecture with an Actor-Critic architecture for optimal testing strategy generation with the patient's representation as the state, and the predicted labels as environmental feedback. The method is experimented on 2 popular EHR datasets and compared with other EHR prediction models.

**Strengths:**

1) Well-designed Reinforcement Learning mechanism: I believe using Actor-Critic with the patient's history representation acting as a state, and the quality of the model's predictions as the environment's feedback. The design of the reward function is concise and logical.
2) Well-written content: The paper, in general, is quite easy to follow, with illustrative figures and a clear explanation of the technical content.
3) State-of-the-art results: This work achieves great results compared with contemporary works, significantly outperforming other baselines on two popular datasets, MIMIC-III and eICU.

**Weaknesses:**

1) Cross-modal alignment mechanism lacks novelty
The cross-modal alignment mechanism lacks originality. The general idea is still projecting cross modal data of an instance to a shared space, and maximize their cosine similarities. Well-established works [1], [2], [3] have operated on the same general idea, with only slight difference in implementation. What are the research gaps of existing cross-modal techniques? How the proposed technique differs and solves the existing research gaps?

2) Require more rigorous experimental comparisons.
While the method reports very good accuracy compared with other state-of-the-art models, I believe a more comprehensive evaluation will be more reliable and conclusive for the actual performance of the model. The paper only includes a single clinical task, specifically Heart Failure Prediction. However, in real-world use cases, many other clinically important prediction tasks may require a more in-depth analysis of the patient's health history, such as mortality prediction, readmission prediction, or prediction of other critical conditions, like COPD. Also, the results can vary according to the different patient split, or simply a random seed set during the training process. Reporting the results over multiple runs with statistical tests will make the conclusions more reliable and generalizable.

3) No ablation studies: The paper mentions an ablation study in Table 3 (424-431), but that content is not present in the paper at all (Table 3 is actually a hyperparameter table). As such, the effectiveness of each proposed component cannot be justified.

4) Unclarity in the application of LLMs: The paper uses a DeepSeek model to capture temporal trends in textual form. However, the paper does not mention which models (DeepSeek R1, V3?...) are actually used, the prompt used for this task, or other settings such as top-p or top-k. This is very critical information that was left out.

5) Robustness to other pretrained/foundation models: How does this method translate when applying other LLMs for the task of reasoning, especially for lighter-weight models? As of now, the DeepSeek model seems to be a critical point of failure for the model.

6) The problem of hallucination Additionally, how does the paper handle the problem of hallucination, which is prone to occur in specialized medical domains? If the LLM encounters hallucination going undetected, then that could render the lab value process modelling ineffective. The most

7) Computation Cost: The paper utilizes a Large Language Model (possibly hundreds of billions of parameters) for reasoning, then another Pretrained Text Embedding model for representation (hundreds of millions of parameters). Compared with other contemporary works that utilize much more lightweight architectures, is this method applicable for healthcare institutions with limited resources, or use cases that require fast processing time? Is the few percent gap in accuracy justified to implement a much more complex, error-prone, and expensive system?

8) Soundness of the Cross-model alignment mechanism: From what I understand, the alignment mechanism is a regularization function for similarity maximization between text and graph embeddings of the same patient. But does a rigid similarity maximization actually work well for two completely different modalities, encoding different information with vastly different architectures? Especially in the case of a used dataset, where only a small number of a few thousand samples are available, the effectiveness of the proposed method is in doubt. There is no theoretical justification or experimental results to support this proposal presented in this paper (maybe it's in the excluded ablation studies) either.

References:
[1] Radford, A., Kim, J.W., Hallacy, C., Ramesh, A., Goh, G., Agarwal, S., Sastry, G., Askell, A., Mishkin, P., Clark, J. and Krueger, G., 2021, July. Learning transferable visual models from natural language supervision. In International conference on machine learning (pp. 8748-8763). PmLR.
[2] Li, J., Selvaraju, R., Gotmare, A., Joty, S., Xiong, C. and Hoi, S.C.H., 2021. Align before fuse: Vision and language representation learning with momentum distillation. Advances in neural information processing systems, 34, pp.9694-9705.
[3] Zhai, X., Wang, X., Mustafa, B., Steiner, A., Keysers, D., Kolesnikov, A. and Beyer, L., 2022. Lit: Zero-shot transfer with locked-image text tuning. In Proceedings of the IEEE/CVF conference on computer vision and pattern recognition (pp. 18123-18133).

**Questions:**

Please refer to Weaknesses.

---

### Official Review · Reviewer_p9AT · 2025-11-02

**Soundness:** 1
**Presentation:** 2
**Contribution:** 1
**Rating:** 0
**Confidence:** 4

**Summary:**

This paper introduces LLM-HFR-RL, an end-to-end heart failure risk prediction framework integrating LLM for time-series laboratory data summarization, cross-modal alignment of structured and unstructured EHR data, and RL for clinical decision support. This framework converts numerical laboratory time series into semantic summaries, aligns three modalities (laboratory summaries, clinical notes, and structured data) via cosine similarity loss, and employs an Actor-Critic RL agent with multi-objective reward functions to recommend cost-effective laboratory tests.

**Strengths:**

This paper proposes an innovative framework that combines large language models (LLMs) and reinforcement learning (RL) for heart failure risk prediction. The integration of structured lab data, unstructured clinical notes, and LLM-generated summaries through cross-modal alignment is conceptually strong. The RL module extends the model from risk prediction to cost-aware, early-intervention decision support. Experiments on MIMIC-III and eICU show clear performance gains, demonstrating both technical novelty and clinical relevance.

**Weaknesses:**

1. This paper addresses RL methods where EHR definitions are ambiguous, such as State, action, transition, and reward—particularly action decision-making. Additionally, similar approaches already exist in ACL 2025 [1].
2. Theoretical inconsistency exists between PPO and Q-Learning: PPO is an on-policy
policy gradient using data generated solely by the current policy, while Q-Learning is
off-policy convergence-based. However, the authors add these together in Equation 27, constituting a significant error.
3. Table 1 lacks detailed 5-fold cross-validation. It should include the mean and standard deviation for each metric to better highlight your method's performance.
4. Why was DeepSeek chosen as the base framework? Comparisons with other frameworks like Mistral, Qwen, Gemma, etc., are absent.
5. Figure 2 in the paper uses demographic data. Which specific data were utilized? These details require discussion, especially since demographics are important clinical factors for heart failure.

[1] DiaLLMs: EHR Enhanced Clinical Conversational System for Clinical Test Recommendation and Diagnosis Prediction, 2025, ACL

**Questions:**

I recommend that the authors provide a clearer description of the model architecture and elaborate on the motivation for adopting RL, referencing prior works that have explored similar directions.

**Details Of Ethics Concerns:**

Dear ICLR Committee,

I reviewed the same paper for AAAI 2026, where it was rejected. The current submission to ICLR2026 only adds an RL method, but most of the experimental results remain unchanged. This raises concerns about potential experimental falsification. I am reporting this for your attention.

---

### Official Review · Reviewer_kB6h · 2025-11-12

**Soundness:** 2
**Presentation:** 2
**Contribution:** 2
**Rating:** 2
**Confidence:** 3

**Summary:**

This paper proposes LLM-HFR-RL, a framework for heart failure risk (HFR) prediction that integrates three key modules: (1) LLM-driven longitudinal laboratory index summarization that transforms discrete test sequences into textual trend summaries, (2) a ternary cross-modal alignment architecture integrating structured test sequences, LLM-generated summaries, and clinical text, and (3) a reinforcement learning (RL) decision engine using PPO to learn optimal testing strategies via a multi-objective reward function. The framework is evaluated on MIMIC-III and eICU datasets, showing improvements over baseline methods in AUC, F1-score, and accuracy for HF prediction.

**Strengths:**

- The paper addresses important clinical problem: Balancing accurate heart failure prediction with cost-effective test selection is practically relevant
- The framework integrates multiple data modalities (lab tests, clinical notes, structured EHR) and bridges prediction with decision-making
- Testing on both MIMIC-III and eICU provides some evidence of generalization across ICU settings

**Weaknesses:**

- No comparison showing LLM-generated summaries outperform simpler temporal encoding methods
- The cross-modal alignment poorly justified: No comparison with other multimodal fusion methods
- Limited information about joint training.
- State representation is a simple linear projection. This needs justification why it is not too simplistic.
- No analysis on learned policies

**Questions:**

- can you provide clinical validation that LLM-generated trend summaries are accurate ?
- can you analyze the learned testing policies?
- can you provide details on joint optimization? e.g.m how are gradients balanced?
- how LLM component plays role  (LLM summaries vs. raw numerical sequences)?

---

### Meta-Review · Area_Chair_bask · 2026-01-10

**Summary:**

The paper was evaluated as below the acceptance threshold, as all reviewers assigned low scores. While reviewers agree that the clinical problem is important and that the system is competently engineered, the submission falls short due to unresolved concerns regarding methodological clarity, the soundness and motivation of the RL formulation, insufficient experimental rigor, and limited demonstrated novelty. To strengthen the work, the authors should more clearly justify the necessity of each component through targeted ablation studies, simplify or better motivate the RL design, substantially improve the robustness and transparency of the experimental evaluation, and integrate all clarifications directly into the main paper.

**Reviewer Concerns:**

No rebuttal was provided, and thus none of the reviewers’ concerns were addressed.

**Reviewer Scores:**

No rebuttal was provided; as a result, none of the reviewers’ scores were updated.

Reviewer kB6h: Score: 2

Reviewer p9AT: Score: 0

Reviewer QkjM: Score: 2

Reviewer L5KX: Score: 2

---

### Decision · Program_Chairs · 2026-01-26

Reject